# Genomic analysis of Shiga toxin-producing *Escherichia coli* from patients and asymptomatic food handlers in Japan

Hiroaki Baba[1]☯*, Hajime Kanamori[1]☯, Hayami Kudo[2], Yasutoshi Kuroki[2], Seiya Higashi[2], Kentaro Oka[2], Motomichi Takahashi[2], Makiko Yoshida[1], Kengo Oshima[1], Tetsuji Aoyagi[1], Koichi Tokuda[1], Mitsuo Kaku[1]

**1** Department of Infectious Diseases, Internal Medicine, Tohoku University Graduate School of Medicine, Sendai, Miyagi, Japan, **2** Miyarisan Pharmaceutical Co., Ltd., Saitama-shi, Saitama, Japan

☯ These authors contributed equally to this work.
\* hbaba48@med.tohoku.ac.jp

**Data Availability Statement:** All of whole-genome sequence data reported within the paper are available from the DDBJ/EMBL/Genbank Sequence

## Abstract

Shiga toxin-producing *Escherichia coli* (STEC) can cause severe gastrointestinal disease and colonization among food handlers. In Japan, STEC infection is a notifiable disease, and food handlers are required to undergo routine stool examination for STEC. However, the molecular epidemiology of STEC is not entirely known. We investigated the genomic characteristics of STEC from patients and asymptomatic food handlers in Miyagi Prefecture, Japan. Whole-genome sequencing (WGS) was performed on 65 STEC isolates obtained from 38 patients and 27 food handlers by public health surveillance in Miyagi Prefecture between April 2016 and March 2017. Isolates of O157:H7 ST11 and O26:H11 ST21 were predominant (n = 19, 29%, respectively). Non-O157 isolates accounted for 69% (n = 45) of all isolates. Among 48 isolates with serotypes found in the patients (serotype O157:H7 and 5 non-O157 serotypes, O26:H11, O103:H2, O103:H8, O121:H19 and O145:H28), adhesion genes *eae*, *tir*, and *espB*, and type III secretion system genes *espA*, *espJ*, *nleA*, *nleB*, and *nleC* were detected in 41 to 47 isolates (85–98%), whereas isolates with other serotypes found only in food handlers were negative for all of these genes. Non-O157 isolates were especially prevalent among patients younger than 5 years old. Shiga-toxin gene *stx1a*, adhesion gene *efa1*, secretion system genes *espF* and *cif*, and fimbrial gene *lpfA* were significantly more frequent among non-O157 isolates from patients than among O157 isolates from patients. The most prevalent resistance genes among our STEC isolates were aminoglycoside resistance genes, followed by sulfamethoxazole/trimethoprim resistance genes. WGS revealed that 20 isolates were divided into 9 indistinguishable core genomes (<5 SNPs), demonstrating clonal expansion of these STEC strains in our region, including an O26:H11 strain with *stx1a*+*stx2a*. Non-O157 STEC with multiple virulence genes were prevalent among both patients and food handlers in our region of Japan, highlighting the importance of monitoring the genomic characteristics of STEC.

Read Archive (SRA) (accession numbers DRX149942 to DRX150006).

**Funding:** All named authors have approved this manuscript and have no conflict of interest. The funder, Miyarisan Pharmaceutical Co., Ltd. provided support in the form of salaries for authors [HK, YK, SH, KO, MT], but did not have any additional role in the study design, data collection and analysis, decision to publish, or preparation of the manuscript. The specific roles of these authors are articulated in the 'author contributions' section.

**Competing interests:** The authors have declared that no competing interests exist. This does not alter our adherence to PLOS ONE policies on sharing data and materials.

## Introduction

Shiga toxin-producing *Escherichia coli* (STEC) cause various gastrointestinal diseases in humans, including life-threatening hemolytic uremic syndrome (HUS) [1]. Although O157: H7 is the predominant pathogenic serotype, severe infections caused by non-O157 serogroups are increasingly reported worldwide [2]. STEC transmission occurs through intake of contaminated food or via person-to-person spread, with large-scale outbreaks having been reported [1]. In Japan, food safety control measures and a STEC surveillance system were instituted after a massive STEC epidemic occurred in Sakai city in 1996, and STEC infection became a notifiable disease [3]. To prevent the spread of infection via food, the Japanese Ministry of Health, Labor and Welfare requires food handlers to undergo routine stool examination for various infectious pathogens, including STEC, and asymptomatic STEC carriers are legally restricted from working as food handlers [4]. Despite these efforts, approximately 4,000 cases of STEC infection are still reported annually in Japan [3].

Shiga toxin (Stx) is the most important STEC virulence factor. Stx has 2 subtypes with variants, which are Stx1 (*stx1a*, *stx1c*, and *stx1d*) and Stx2 (*stx2a*, *stx2b*, *stx2c*, *stx2d*, *stx2e*, *stx2f*, and *stx2g*) [5]. In addition, highly pathogenic STEC possess other virulence factors that include adhesins, other toxins, and protein secretion systems [1]. Detection of genes encoding these virulence factors in STEC strains could provide useful information about risk factors that may contribute to human disease. In recent years, there has been a worldwide increase of reports about antimicrobial resistance (AMR) among STEC strains [6]. In STEC carriers taking antibiotics, resistant STEC strains may have a selection advantage over other intestinal bacteria. Because of the public health implications of STEC infection, a comprehensive investigation of virulence and AMR factors is required to assess the potential pathogenicity and antibiotic resistance of STEC isolates from patients and asymptomatic food handlers. Some European authors have investigated the molecular characteristics of STEC isolates [7, 8], but no molecular epidemiological studies have been done to assess the relationship between STEC isolates from patients and food handlers. In the present study, we investigated molecular epidemiology of STEC infection in Miyagi Prefecture, Japan, and performed whole-genome sequencing (WGS) to characterize the genomic features of STEC isolates from patients and asymptomatic food handlers including virulence factors and AMR genes.

## Material and methods

### Bacterial strains and clinical data

From April 2016 to March 2017, we collected all 65 epidemiologically unlinked STEC isolates detected through public health surveillance for infectious diseases in Miyagi Prefecture, which is located in central northeastern Japan and has a population of about 2.3 million. Thirty-eight isolates were obtained from fecal samples of hospital patients and 27 isolates were detected by routine stool examination of asymptomatic food handlers. Isolation of STEC from stool samples was done with sorbitol-MacConkey agar containing cefixime and tellurite in addition to conventional *E. coli* isolation agar (e.g., triple sugar iron agar and lysine-indole-motility medium). A latex agglutination test (VTEC-RPLA, Denka Seiken, Japan) and PCR with the EVT-1&2 and EVS-1&2 primers (TaKaRa Biomedicals, Tokyo, Japan) were used to detect Stx and Stx genes, respectively.

Patient data (e.g., age, sex, and clinical manifestation) were also collected through STEC public health surveillance. Patients were divided into four age groups: infants and small children (0–4 y), older children and adolescents (5–19 y), adults (20–64 y), and older people (>65 y) [9]. The age-specific incidence of STEC infections per 100,000 population by age group was calculated using Miyagi Prefecture population data obtained from the National Institute of

Population and Social Security Research website (http://www.ipss.go.jp/). This study was approved by the institutional review board of Tohoku University Graduate School of Medicine (IRB no. 2018-1-368).

## Whole-genome sequencing

Bacterial DNA from the 65 isolates was extracted as described previously [10], and a DNA library was prepared from each sample with a NEBNext Ultra DNA Library Prep Kit for Illumina (New England Biolabs, Ipswich, MA, USA) according to the manufacturer's instructions. Then WGS was performed using a MiSeq (Illumina, San Diego, CA, USA) to generate paired-end 300-bp reads, resulting in an average of 5,556,073 read pairs per isolate. All samples showed a minimum average 30-fold coverage. The passing filter ranged from 90.88 to 96.74% (mean, 93.19%), and the average Q30 ranged from 78.30 to 87.21% (mean, 83.50%). All of the sequence data reported here have been deposited in DDBJ/EMBL/Genbank Sequence Read Archive (SRA) under accession numbers DRX149942 to DRX150006.

## Genetic analysis

Sequence reads were trimmed of adaptors and filtered to remove reads shorter than 36 bp using Trimmomatic [11], followed by assembly using Platanus assembler v 1.2.4. [12]. Specific genes and alleles were identified with the bioinformatic pipeline of the Center for Genomic Epidemiology (http://www.genomicepidemiology.org), using the default setting of a 90% ID threshold and 60% minimum gene length overlap, except where otherwise stated. Specifically, SerotypeFinder 1.1 [13] was used to identify serogenotypes, MLST Finder 2.0 [14] was employed for multilocus sequence typing (MLST), VirulenceFinder 1.5 [15] was used for virulence genes, and ResFinder 3.0 [16] was employed for acquired AMR genes. We also searched for the major adhesin gene *saa*, which cannot be detected by VirulenceFinder, using BLAST (http://blast.ncbi.nlm.nih.gov).

A non-recombinogenic core genome single-nucleotide polymorphism (SNP)-based phylogeny was generated with Parsnp v 1.2 (https://harvest.readthedocs.io/en/latest/content/parsnp/quickstart.html#advanced-usage) [17] using the 65 STEC isolates and the following 6 reference genomes: STEC O157:H7 Sakai (GenBank accession numbers: BA000007), STEC O157:H7 strain EDL933 (AE005174), STEC O157:H7 strain EC4115 (CP001164), STEC O157:H7 strain TW14359 (CP001368), STEC O26:H11 strain 11368 (AP010953), and STEC O103:H2 strain 12009 (AP010958). Differences in the number of SNPs between STEC strains were calculated by Parsnp. Clonal STEC strains were defined as isolates with less than 5 SNPs [18]. A phylogenetic tree with 1000 boostrap replicates was constructed by using the randomized accelerated maximum likelihood (RAxML) program v 8.2.12 (https://cme.h-its.org/exelixis/web/software/raxml/index.html) [19], and it was visualized with FigTree (http://tree.bio.ed.ac.uk/software/figtree/).

## Statistical analysis

Fisher's exact test and the two-sample t-test were used for analysis of categorical variables and continuous variables, respectively. In all analyses, $P < 0.05$ was considered statistically significant.

# Results

## Serogenotyping and MLST

Among the 65 STEC isolates, serogenotyping and MLST revealed 18 different serogroups and 20 different sequence types (Fig 1). Seven sequence types had already been reported as STECs

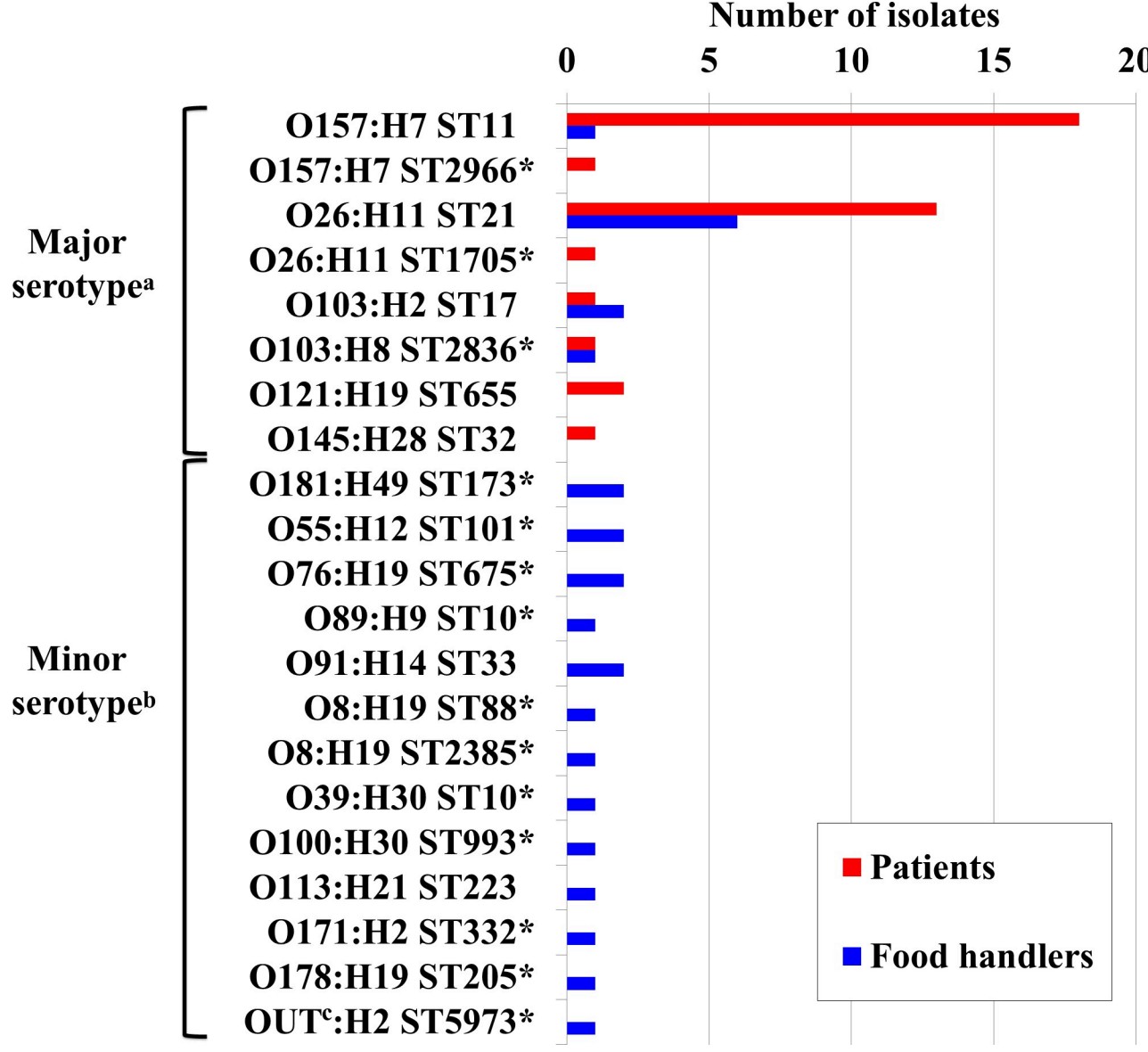

**Fig 1. Frequency of O/H serotypes and sequence types (STs) among Shiga-toxin producing *Escherichia coli* (STEC) isolates.** Red and blue bars represent the number of isolates from patients and asymptomatic food handlers, respectively. *New STEC STs. [a]Major serotypes: serotypes found in patients in this study. [b]Minor serotypes: serotypes only found in food handlers in this study. [c]OUT: O-serotype untypable.

causing human disease according to Enterobase (http://enterobase.warwick.ac.uk), while the other 13 sequence types, including O103:H8 ST2836, are new as STEC strains. An O-group was not detected in the one OUT (O-serotype untypable) isolate by SerotypeFinder. In this isolate, no O-processing genes (*wzx*, *wzy*, *wzm*, *wzt*) were detected by BLAST. It is possible that this isolate could be assigned to serogroups O14 or O57 since O-processing genes for these serogroups have not been found in their genomes [20], or it could represent a new serogroup.

Non-O157 isolates accounted for 69% (n = 45) of all isolates. O157 isolates were significantly more frequent among the patients than the food handlers (19/38 isolates from patients versus 1/27 isolates from food handlers, *P*<0.001), whereas non-O157 isolates were significantly more frequent among the food handlers. Among isolates from the patients, the

predominant serotype was O157:H7 (n = 19, 50%) followed by O26:H11 (n = 14, 37%), while O26:H11 was predominant (n = 6, 22%) among isolates from the food handlers, followed by O103:H2, O181:H49, O55:H12, O76:H19, O91:H14, and O8:H19 (n = 2, 7.4%, respectively). Isolates from patients belonged to the following 6 serotypes (O157:H7 and 5 non-O157 serotypes, which were O26:H11, O103:H2, O103:H8, O121:H19, and O145:H28) (hereinafter called "major serotypes"), while the remaining 12 serotypes were only found in the food handlers (hereinafter called "minor -serotypes") (Fig 1). Isolates with the same serotype generally belonged to the same sequence type, except that O157:H7 isolates belonged to ST11 and ST2966, O26:H11 isolates belonged to ST21 and ST1705, and O8:H19 isolates belonged to ST88 and ST2385. There was only one SNP difference between ST11 and ST2966 in *purA*, as well as between ST21 and ST1705 in *gyrA*, suggesting a close relation between these sequence types.

## Clinical characteristics of patients and food handlers

The majority of the patients (28/38, 74%) had bloody diarrhea and 1 patient (2.6%) developed HUS, while the symptoms of the remaining 10 patients (36%) were unknown. The patients were aged between 11 months and 85 years (median: 17.5 years), whereas the food handlers were aged from 21 to 71 years (median: 54 years). Sixty-one percent (15/23) of the patients and 74% (20/27) of the food handlers were female.

The annual age-specific incidence of O157 and non-O157 infections in Miyagi Prefecture during the study period is summarized in Table 1. In this region, the overall incidence of STEC infection was 1.6, with infants and small children having the highest incidence (13.5, $P<0.001$ vs. each other age group). Importantly, the incidence of non-O157 infections was significantly higher in infants and small children (11.3) than in the other age groups ($P<0.001$ vs. each other age group), while there was no significant difference in the incidence of O157 infections among the age groups.

## Virulence genes

Among a total of 76 virulence genes registered in VirulenceFinder, 44 genes (58%) were detected among the isolates and there was a median of 17 virulence genes per isolate (range: 1–26). Isolates from the patients harbored significantly more virulence genes than isolates from the food handlers (a median of 18 and 10 virulence genes per isolate, respectively, $P<0.001$), and O157 isolates had significantly more virulence genes than non-O157 isolates (a median of 19 and 16 virulence genes per isolate, respectively, $P = 0.013$). Eight different Stx subtypes (combinations) were detected among the isolates, with *stx1a*-only being most frequent (n = 27, 42%), followed by *stx1a*+*stx2a* (n = 12, 18%). The distribution of virulence genes among isolates from the patients or food handlers and among O157 or non-O157 isolates is shown in S1 Fig and S1 Table.

**Table 1.  Annual age-specific incidence of Shiga toxin-producing *Escherichia coli* (STEC) infection in Miyagi Prefecture during the study period.**

| Age group | Population (n) | No. of cases | | | Incidence per 100,000 population | | |
|---|---|---|---|---|---|---|---|
| | | All STEC infections | O157 | non-O157 | All STEC infections | O157 | non-O157 |
| Infants and small children | 88,787 | 12 | 2 | 10 | 13.5 | 2.3 | 11.3 |
| Older children and adolescents | 311,185 | 8 | 3 | 5 | 2.6 | 1.0 | 1.6 |
| Adults | 1,296,353 | 11 | 8 | 3 | 0.9 | 0.6 | 0.2 |
| Older people | 588,240 | 7 | 6 | 1 | 1.2 | 1.0 | 0.2 |
| Total | 2,333,899 | 38 | 19 | 19 | 1.6 | 0.8 | 0.8 |

Isolates with major serotypes had significantly more virulence genes than isolates with minor serotypes (a median of 18 and 7 virulence genes per isolate, respectively, $P<0.001$). The Stx subtype *stx1a* was significantly more frequent among isolates with major serotypes, while *stx2d* and *stx2e* were detected significantly more often in isolates with minor serotypes. Among the 48 isolates with major serotypes, adhesion genes *eae*, *tir*, and *espB* were detected in 47 (98%), 46 (96%), and 45 (94%) isolates respectively, and secretion system genes *espA*, *espJ*, *nleA*, *nleB*, and *nleC* were detected in 46 (96%), 46 (96%), 41 (85%), 45 (94%), and 41 (85%) isolates respectively, whereas all isolates with minor serotypes were negative for all of these genes (all $P<0.001$) (Table 2).

Among the isolates from patients, O157 and non-O157 isolates had a comparable number of virulence genes (a median of 17 virulence genes per isolate, respectively, $P = 0.32$). Stx gene *stx1a*, adhesion gene *efa1*, secretion system genes *espF* and *cif*, and fimbrial gene *lpfA* were significantly more frequent among non-O157 isolates from patients than among O157 isolates from patients (Table 2). In addition, *stx1a*, *lpfA*, and secretion system gene *tccP* were significantly more frequent in isolates from patients that were infants and small children than in isolates from patients of other age groups (S2 Table). The isolate from the 1 patient with HUS had *stx2c*-only, as well as adhesion genes *eae*, *tir*, and *espB*, and secretion system genes *espA*, *espJ*, *nleA*, *nleB* and *nleC*.

The additional search for *saa* using BLAST showed that only 5 out of 65 isolates (7%) possessed this gene. All of these isolates were non-O157 and from food handlers.

## AMR genes

WGS analysis identified 20 acquired AMR genes in 18 STEC isolates (28% of all 65 STEC isolates) (S1 Fig). The β-lactamase gene ($bla_{TEM-1B}$) was detected in 7 isolates (11%). There were 14 isolates (22%), 16 isolates (25%), 11 isolates (17%), 3 isolates (5%), and 2 isolates (3%) with at least one of the sulfamethoxazole/trimethoprim, aminoglycoside, tetracycline, macrolide, and phenicol resistance genes, respectively. The distribution of AMR genes was similar among isolates from the patients or food handlers and among O157 or non-O157 isolates. Aminoglycoside resistance genes were less frequent among isolates with major serotypes than isolates with minor serotypes (22/48 isolates with major serotypes versus 16/17 isolates with minor serotypes, $P<0.001$).

## Phylogenetic analysis

Phylogenetic analysis was performed using 132,711 SNPs identified within the core genome of 71 STEC isolates (including the 6 reference strains) (Fig 2). The STEC isolates were divided into two clades, O157 and non-O157, except that the O145:H28 isolate clustered with the O157 isolates. Isolates with the same O serotype formed a cluster together, except for O103 and O8. In addition, the O103:H8 ST2836 isolates clustered with the O26:H11 isolates and were separated from the O103:H2 ST17 isolate. Within the O157:H7 cluster, isolates positive for *stx1a+stx2a* formed a subcluster. Isolates positive for microcin genes *mcmA*, *mchB*, *mchC*, and *mchF* were assigned to a subcluster within the O26:H11 cluster.

In pairwise comparisons, the median number of SNP differences between different core genomes was 423 within O157 (range, 0 to 584), compared to 22,054 (range, 0 to 54,528) for non-O157 genomes overall. Twenty isolates (6, 10, 2, and 2 isolates with O157:H7 ST11, O26: H11 ST21, O103:H8 ST2836, and O76:H19 ST675, respectively) were divided into 9 indistinguishable core genomes ($<5$ SNPs) (Fig 2). Among these 9 core genomes, 5 were from patients, 3 were from food handlers, and one was isolated from both a patient and a food

**Table 2. Distribution of putative virulence genes among Shiga toxin-producing *Escherichia coli* (STEC) isolates with major/minor serotypes and O157/non-O157 isolates from patients.**

| | | No. of isolates (%) | | | | | |
|---|---|---|---|---|---|---|---|
| | | Major serotype (n = 48) | Minor serotype (n = 17) | *P* | O157 from patients (n = 19) | Non-O157 from patients (n = 19) | *P* |
| Shiga-toxin pattern | *stx1a* | 24 (50) | 3 (18) | 0.024 | 0 | 15 (79) | <0.001 |
| | *stx1c* | 0 | 2 (12) | | 0 | 0 | |
| | *stx2a* | 3 (6) | 2 (12) | | 1 (5) | 2 (11) | |
| | *stx2c* | 8 (17) | 1 (6) | | 8 (42) | 0 | 0.003 |
| | *stx2d* | 0 | 3 (18) | 0.016 | 0 | 0 | |
| | *stx2e* | 0 | 4 (24) | 0.015 | 0 | 0 | |
| | *stx1a+stx2a* | 11 (23) | 1 (6) | | 9 (47) | 2 (11) | 0.029 |
| | *stx1a+stx2c* | 1 (2) | 0 | | 0 | 0 | |
| Adhesins | *eae* | 47 (98) | 0 | <0.001 | 19 (100) | 19 (100) | |
| | *tir* | 46 (96) | 0 | <0.001 | 18 (95) | 19 (100) | |
| | *espB* | 45 (94) | 0 | <0.001 | 19 (100) | 18 (95) | |
| | *iha* | 32 (67) | 8 (47) | | 18 (95) | 5 (26) | <0.001 |
| | *efa1* | 21 (44) | 0 | <0.001 | 0 | 14 (74) | <0.001 |
| Toxins | *ehxA* | 44 (92) | 8 (47) | <0.001 | 18 (95) | 17 (89) | |
| | *toxB* | 37 (77) | 0 | <0.001 | 18 (95) | 12 (63) | 0.042 |
| | *astA* | 41 (85) | 2 (12) | <0.001 | 19 (100) | 15 (79) | |
| | *subA* | 0 | 8 (47) | <0.001 | 0 | 0 | |
| | *cdtB* | 0 | 3 (18) | 0.016 | 0 | 0 | |
| | *sta1* | 0 | 1 (6) | | 0 | 0 | |
| | *senB* | 0 | 2 (12) | | 0 | 0 | |
| Secretion system | *espA* | 46 (96) | 0 | <0.001 | 18 (95) | 19 (100) | |
| | *espF* | 28 (58) | 0 | <0.001 | 5 (26) | 14 (74) | 0.009 |
| | *espI* | 2 (4) | 0 | | 0 | 2 (11) | |
| | *espJ* | 46 (96) | 0 | <0.001 | 18 (95) | 18 (95) | |
| | *nleA* | 41 (85) | 0 | <0.001 | 19 (100) | 13 (68) | 0.020 |
| | *nleB* | 45 (96) | 0 | <0.001 | 19 (100) | 19 (100) | |
| | *nleC* | 41 (85) | 0 | <0.001 | 18 (95) | 14 (74) | |
| | *etpD* | 19 (40) | 0 | 0.001 | 17 (89) | 0 | <0.001 |
| | *cif* | 23 (48) | 0 | <0.001 | 0 | 16 (84) | <0.001 |
| | *tccP* | 23 (48) | 0 | <0.001 | 2 (11) | 8 (42) | |
| SPATEs[a] | *espP* | 43 (90) | 7 (41) | <0.001 | 19 (100) | 17 (89) | |
| | *pic* | 1 (2) | 2 (12) | | 1 (5) | 0 | |
| | *sepA* | 0 | 1 (6) | | 0 | 0 | |
| Colicins | *cma* | 1 (2) | 3 (18) | | 1 (5) | 0 | |
| | *cba* | 16 (33) | 4 (24) | | 3 (16) | 5 (26) | |
| | *celb* | 3 (6) | 4 (24) | | 0 | 3 (16) | |
| Microcins | *mcmA* | 4 (8) | 2 (12) | | 0 | 4 (21) | |
| | *mchB* | 4 (8) | 2 (12) | | 0 | 4 (21) | |
| | *mchC* | 4 (8) | 2 (12) | | 0 | 4 (21) | |
| | *mchF* | 4 (8) | 2 (12) | | 0 | 4 (21) | |
| Others | *lpfA* | 25 (52) | 14 (82) | 0.043 | 1 (5) | 17 (89) | <0.001 |
| | *katP* | 41 (85) | 2 (12) | <0.001 | 19 (100) | 15 (79) | |
| | *ireA* | 0 | 5 (29) | 0.000 | 0 | 0 | |
| | *gad* | 20 (42) | 2 (12) | 0.036 | 11 (58) | 1 (5) | 0.001 |
| | *iss* | 46 (96) | 14 (82) | | 18 (95) | 18 (95) | |
| | *CapU* | 0 | 1 (6) | | 0 | 0 | |

[a]SPATE: Serine protease autotransporters of *Enterobacteriaceae*. *P* values are only shown if *P*<0.05.

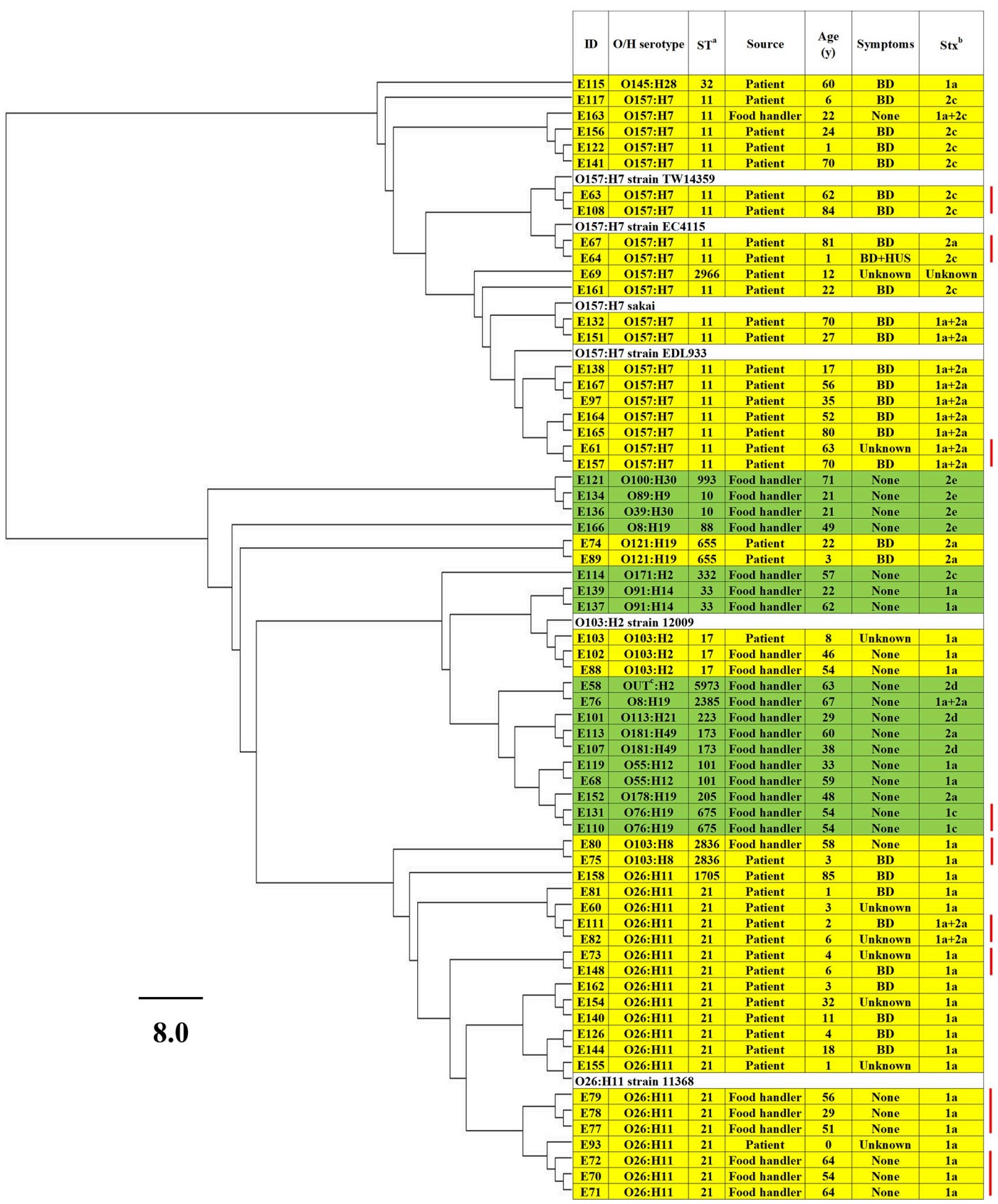

**Fig 2. Maximum likelihood tree based on the core genome shared by Shiga toxin-producing *Escherichia coli* (STEC) isolates, including reference strains.** Scale bar represents 8.0 nucleotide substitutions per site. A total of 132,711 SNPs were identified in the core genome. The average number of SNPs showing differences between each pair of STEC isolates was 33,603. Yellow shading = isolates with major serotypes; light green shading = isolates with minor serotypes. Closely related isolates (SNP differences <5) are highlighted with red lines behind them. [a]ST: sequence type. [b]Stx: Shiga toxin. [c]OUT: O-serotype untypable.

handler. Two of the 4 O26:H11 strains, including one strain positive for *stx1a+stx2a*, were from patients under 5 years of age.

## Discussion

To the best of our knowledge, this is the first genomic epidemiological study to investigate STEC isolates from patients and asymptomatic food handlers. Isolates with 5 non-O157 serotypes (O26:H11, O103:H2, O103:H8, O121:H19, and O145:H28), which had as many virulence genes as O157 isolates were prevalent among both patients and food handlers, whereas isolates with the remaining 12 serotypes were only found in food handlers and were negative for major virulence genes, *eae*, *tir*, *espB*, *espA*, *espJ*, *nleA*, *nleB*, and *nleC*. Non-O157 isolates were especially prevalent among children under 5 years of age. WGS analysis revealed clonal expansion of highly virulent STEC strains (e.g., O26:H11 strain with *stx1a+stx2a*) in our region (Miyagi Prefecture, Japan).

Although the overall isolation rate of non-O157 STEC strains was reported to be 30–40% in Japan [3], more than half of the STEC isolates were non-O157 strains in our region. Similar to the increment of non-O157 isolates revealed by this study, non-O157 infections have been increasingly reported worldwide [2, 21]. Notably, non-O157 infection was predominant among children under 5 years old in this study. Among the isolates from patients, adhesin gene *efa1*, secretion system genes *espF* and *cif*, and the gene *lpfA* encoding fimbriae, were significantly more frequent among non-O157 isolates than O157 isolates. *lpfA* was reported to be involved in prolonged shedding of STEC in young children [22]. In Japan, direct person-to-person contact is the suspected route of transmission for the majority of STEC infection outbreaks among children [23]. Prolonged shedding of STEC can facilitate its spread. Our findings suggested that these genes may be associated with a high frequency of non-O157 STEC infection among children.

The isolates with major serotypes (O157:H7, O26:H11, O103:H2, O103:H8, O121:H19, and O145:H28) harbored significantly more virulence genes than the isolates with minor serotypes. Apart from O103:H8, these major serotypes have been linked to epidemics and serious infections and have been frequently detected among clinical isolates worldwide [24]. Most of the isolates with major serotypes, including the isolate from a patient with HUS, possessed adhesion genes *eae*, *tir*, and *espB*, and secretion system genes *espA*, *espJ*, *nleA*, *nleB* and *nleC*, whereas the isolates with minor serotypes were negative for all of these genes. Studies have shown that the *eae* gene encoding intimin, an outer membrane protein involved in close attachment, is closely linked to the pathogenesis of STEC infection, along with other genes clustering on the bacterial chromosome (such as *tir*, *espA*, *espB*, and *espJ*) that form a pathogenicity island called the locus of enterocyte effacement [8, 25]. Other studies have shown that effectors outside this locus encoded by *nleA*, *nleB*, and *nleC* are required to form attaching and effacing lesions in the intestinal epithelium, which allow STEC to colonize the human gut [26]. Accordingly, these virulence factors may play a key role in the pathogenesis of STEC infections. The current STEC surveillance system for food handlers in Japan is only based on serotyping and detection of Stx [3]. However, we think that STEC surveillance should focus on the above-mentioned virulence genes, such as *eae*, *tir*, *espB*, *espA*, *espJ*, *nleA*, *nleB* and *nleC*.

O103:H8 ST2836 STEC with multiple virulence genes was newly detected in this study. O103:H8 ST2836 isolates formed a separate cluster from the known isolates of O103:H2 ST17

on the phylogenetic tree, suggesting that these two clusters of serogroup O103 had different origins. As previously reported, STEC are *E. coli* strains of different lineages that have acquired virulence genes independently at different time points [27], and STEC strains from the same O serogroups are polyphyletic since horizontal transfer of the O-antigen gene can occur among different *E. coli* strains [28]. These points raise the possibility that new serotypes of highly virulent STEC may emerge.

There have only been a limited number of epidemiological studies on AMR in STEC isolates [6]. The resistance genes with the highest prevalence among our STEC isolates were aminoglycoside resistance genes (e.g., *aadA*, *aph(3')-I*, and *str*), followed by sulfamethoxazole/trimethoprim resistance genes (e.g., *sul* and *dfrA*). In Japan, sulfamethoxazole/trimethoprim and aminoglycosides are antibacterial agents commonly used in domestic animals [29], which are the main reservoir of STEC, and the prevalence of aminoglycoside resistance among STEC isolates from cows in Japan has increased during the past decade [30]. Antimicrobial therapy is generally not recommended for STEC infection due to the possible risk of HUS, but it may be beneficial for patients with persistent diarrhea or food handlers with long-term STEC carriage [31]. Transfer of mobile genetic elements was reported to facilitate the spread of AMR genes to other bacteria [6]. Accordingly, it is important to monitor AMR in STEC isolates and prevent misuse/overuse of antibiotics based on the One Health approach [32].

WGS-based phylogenic analysis revealed a variety of SNP variants among isolates from the same serogenotype or same ST clade, suggesting dissemination of diverse STEC strains throughout our region. The O157:H7 isolates with *stx1a+stx2a* formed a subcluster within the O157:H7 cluster, and the O26:H11 isolates positive for microcin genes formed a subcluster in the O26:H11 cluster. STEC isolates possessing *stx1a+stx2a* have been linked to outbreaks associated with a high frequency of HUS [33]. The presence of microcin genes indicates environmental plasticity of the isolates since microcin is a bactericidal antibiotic [34]. These results highlight the fact that diverse strains with differing levels of virulence can exist within the same STEC serogroup.

Our phylogenetic analysis also detected 9 clonal expansions of STEC strains suggesting circulation of these strains among patients and food handlers in our region. One of the strains was serogenotype O26:H11 ST21 strain harboring *stx1a+stx2a*, which differed from a newly emerging virulent O26:H11/H- ST29 STEC clade reported in Japan by Ishijima et al [35]. In general, the majority of STEC O26:H11 isolates are only positive for *stx1a* [36, 37], highly virulent *stx2a*-containing O26:H11 strains have been increasingly reported worldwide in recent years [36]. The spread of O26:H11 strains with *stx2a* could pose a threat in our region. WGS has been employed to investigate the molecular epidemiology of STEC [7], since it is a powerful tool for performing high-resolution molecular typing, population structure analysis, and detailed molecular characterization of microbes [38]. Further genome-based epidemiological studies are needed to provide a better understanding of STEC isolates for assistance in developing prevention and control strategies.

This study had several limitations. First, there were only a few of STEC strains from the same lineage or serotype, although we assessed all of the STEC isolates detected through public health surveillance in our region during the study period. Second, we could only obtain restricted epidemiological and clinical information. Third, while this *in silico* study was focused on putative virulence genes, the pathogenicity of STEC isolates needs to be clarified by *in vitro* and *in vivo* experimental studies.

In conclusion, we found that genetically diverse non-O157 isolates (O26:H11, O103:H2, O103:H8, O121:H19, and O145:H28) with as many important virulence genes as O157 isolates (including *eae*, *tir*, *espB*, *espA*, *espJ*, *nleA*, *nleB* and *nleC*) plus AMR genes (such as aminoglycoside and sulfamethoxazole/trimethoprim resistance genes) were prevalent among both patients

and asymptomatic food handlers in Miyagi Prefecture, Japan. Our WGS analysis demonstrated the importance of monitoring the genomic characteristics of STEC isolates from asymptomatic food handlers in addition to symptomatic patients.

## Supporting information

**S1 Fig. Characteristics and virulence/antimicrobial resistance (AMR) gene profiles of Shiga toxin-producing *Escherichia coli* (STEC) isolates.** Yellow shading = isolates with major serotypes; light green shading = isolates with minor serotypes. The presence (black) or absence (white) of virulence genes and AMR genes is shown.
[a]ST: sequence type. [b]SPATE: Serine protease autotransporters of *Enterobacteriaceae*. [c]OUT: O-serotype untypable.
(TIF)

**S1 Table. Distribution of putative virulence genes among Shiga toxin-producing *Escherichia coli* (STEC) isolates from patients/food handlers and O157/non-O157 isolates.**
[a]SPATE: Serine protease autotransporters of *Enterobacteriaceae*. *P* values are shown only if *P*<0.05.
(XLSX)

**S2 Table. Distribution of putative virulence genes among Shiga toxin-producing *Escherichia coli* (STEC) isolates from patients of infants and small children/patients of different age groups.** [a]SPATE: Serine protease autotransporters of *Enterobacteriaceae*. *P* values are shown only if *P*<0.05.
(XLSX)

## Acknowledgments

We thank Yumiko Takei for her technical help.

## Author Contributions

**Conceptualization:** Hiroaki Baba, Hajime Kanamori, Kentaro Oka, Motomichi Takahashi, Makiko Yoshida, Mitsuo Kaku.

**Data curation:** Hiroaki Baba, Hajime Kanamori.

**Formal analysis:** Hiroaki Baba, Hajime Kanamori, Hayami Kudo, Seiya Higashi, Kentaro Oka, Motomichi Takahashi.

**Investigation:** Hiroaki Baba, Hajime Kanamori, Hayami Kudo, Yasutoshi Kuroki.

**Methodology:** Hiroaki Baba, Hajime Kanamori, Hayami Kudo, Yasutoshi Kuroki, Kentaro Oka, Motomichi Takahashi.

**Project administration:** Hiroaki Baba, Hajime Kanamori.

**Resources:** Makiko Yoshida, Mitsuo Kaku.

**Software:** Seiya Higashi.

**Supervision:** Hiroaki Baba, Hajime Kanamori.

**Validation:** Hiroaki Baba, Hajime Kanamori.

**Visualization:** Hiroaki Baba.

**Writing – original draft:** Hiroaki Baba.

**Writing – review & editing:** Hiroaki Baba, Hajime Kanamori, Makiko Yoshida, Kengo Oshima, Tetsuji Aoyagi, Koichi Tokuda, Mitsuo Kaku.

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
