## [Decision Letter · Decision Letter 0]

5 Sep 2019

PONE-D-19-20606

Genomic analysis of Shiga toxin-producing Escherichia coli from patients and asymptomatic food handlers in Japan

PLOS ONE

Dear Dr. Baba,

Thank you for submitting your manuscript to PLOS ONE. After careful consideration, we feel that it has merit but does not fully meet PLOS ONE’s publication criteria as it currently stands. Therefore, we invite you to submit a revised version of the manuscript that addresses the points raised during the review process.

Please address all of the comments of the two reviewers, including adding bootstrap analysis, criteria for accepting a WGS run, and details about genome assembly (reviewer #2), and please carefully address all of the comments of reviewer #1.  The term non-patient isolates is not entirely clear.  The authors should reconsider using this term or explain it more clearly.  As reviewer #1 pointed out, some of these serotypes have been isolated from patients.

Furthermore, there are a number of errors in English usage and some clarifications are needed.  For example, Line 69:  name the location in Japan.  Lines 78-80:  Where were the patient fecal samples obtained (e.g., hospitals?) and how were the asymptomatic individuals selected?  Please clarify in the text.  Line 91:  Bacterial DNA from the 65 isolates was extracted…..   Line 179:  Again, here the phrase “isolates with patient serotypes” is awkward/unclear.  Also, on line 180:  What is meant by “detected in 41-47 isolates”?  Lines 183-184:  Consider changing this title to read more clearly.  Line 242:  Give examples/information in the text on which highly virulent STEC showed clonal expansion.

And finally, please make the following these changes in the text: 

Line 21:  ….However, the molecular…..Line 23:  ….and asymptomatic food handlers…..Line 30:  Delete “which”Line 39:  …genes were…..Line 49:  ….after a massive…..Line 58:  ….In addition, highly pathogenic STEC contain…..Line 64:  ….a comprehensive….Line 94:  MiSeq…….300-bp reads,…..Line 146:  …..suggesting a close relationship between……Line 151:  …..of the remaining….Line 152:  …..whereas the ages of the food handlers…Line 188:  ….had a comparable….Lines 192-193:  Should this read. …..often found in isolates from patients that were infants and small children….”Line 194:  …..from an HUS patient….Line 205:  ….with patient serotypes…..Line 233:  …patients younger than 5 years of age.Line 247:  …..isolates from patients,….Line 249:  …..present among non-O157 isolates than from O157.Line 270:  The majority (17…..Line 271:  ….low virulence non-patient serotypes.Line 272:  ……focused on patient-related serotypes….Line 276:  ……two clusters of serogroup O103 had different…..Line 279:  …..O-antigen gene…..Line 292:  ….antibiotics based on the One……Line 297:  …..cluster, and the O26:H11 isolates…….

We would appreciate receiving your revised manuscript by Oct 20 2019 11:59PM. To enhance the reproducibility of your results, we recommend that if applicable you deposit your laboratory protocols in protocols.io, where a protocol can be assigned its own identifier (DOI) such that it can be cited independently in the future. For instructions see: http://journals.plos.org/plosone/s/submission-guidelines#loc-laboratory-protocols

We look forward to receiving your revised manuscript.

Kind regards,

Pina Fratamico, Ph.D.

Academic Editor

PLOS ONE

Journal Requirements:

1. Thank you for including your competing interests statement; "The authors have declared that no competing interests exist."

We note that one or more of the authors are employed by a commercial company: Miyarisan Pharmaceutical Co., Ltd.

Reviewers' comments:

Reviewer's Responses to Questions

**Comments to the Author**

1. Is the manuscript technically sound, and do the data support the conclusions?

Reviewer #1: Partly

Reviewer #2: Yes

2. Has the statistical analysis been performed appropriately and rigorously? 

Reviewer #1: Yes

Reviewer #2: Yes

3. Have the authors made all data underlying the findings in their manuscript fully available?

Reviewer #1: Yes

Reviewer #2: Yes

4. Is the manuscript presented in an intelligible fashion and written in standard English?

Reviewer #1: No

Reviewer #2: Yes

5. Review Comments to the Author

Reviewer #1: Major comments

In this study, the authors revealed the characteristics and phylogeny of STEC in genomic level. As scarce information is available for Japanese STEC isolates, the information is useful. However, some of the discussion is superficial and the authors should consider the results in deeper depth. For instance, when the results were compared to the previous surveillance results of Japan or other countries, what is the implication?

In addition, some sentences are difficult to intuitively understand. English editing is recommended if it was not done yet.

Specific comments

Line 23. our region -> Miyagi prefecture, Japan

Line 26. What “n=19, 29%” meant was unclear. There are 20 isolates of O157 and O26 according to Fig. S1.

Line 29. Add “type III” before “secretion system”.

Line 76. remove “During a one-year period”

Line 76. How were strains isolated? Please add the details of isolation.

Are there any epidemiological links between the isolates used in this study? Discuss it in the Discussion section.

Were all the strains isolated in Miyagi prefecture during the study period included in this study?

In this study, serotypes of the isolates were mentioned. However, it is not mentioned how they were determined. For instance, what did OUT or H- mean. No identical O genotype even in in silico analysis? Did H- mean “no motility”? If so, did you search fliC genotype?

Line 94. Add reagent kit name.

Line 97. It seems that the data is not published yet. Please make the data public when the manuscript is submitted.

Line 106. It is useful to search saa sequence, which is the major adhesin in LEE negative STEC. In Virulence Finder, saa cannot be detected, because almost half of the sequence is repeat sequences.

Line 108. What strain was used as reference?

Line 115. Please add the citation of RAxML, the model used, and the number of bootstrap replicates.

Did you remove recombinogenic region?

Line 127. What does “new” mean in this sentence? For instance, several strains of ST2836 of O103:H8 can be found in EnteroBase.

Line 142. According to the surveillance information of Japan, the “non-patient serotypes” in this study have been reported from the patients.

Line 146. How many SNPs are there in the different allele?

Line 158-160. Are they multiple comparison? If so, how was the P values were adjusted?

Line 199. Please remove “and all 18 isolates had at least one AMR gene”, because it is redundant.

Line 214. Is O26:H11 correct, rather than O157:H7?

Line 214. The O26 cluster in unclear in Fig. 1.

L230-233. The meaning of these sentences is unclear. The authors should clarify the implication of the information.

Line 240. What are major virulence genes? LEE-related genes?

Line 241. The etiologic agent of most of the young children patients was O26. Therefore, the serotype should be focused on.

Line 254-256. These are paradoxical statements. The fact “women are more likely to work in food related occupations” do not explain that more than half of the cases of food handlers are women.

Line 272. This suggestion is unclear. Please specify the recommendation (e.g. isolation method).

In Japan (and other countries), “non-patient serotypes” in this study are responsible for severe diseases, although the incidence rate is not high. How do the authors evaluate them?

Line 290. “under certain conditions” should be clarified.

Line 294-298. Subclusters of O157 is also unclear in Fig. 1. What about clade of O157?

Line 297-301. Virulence of O26:H11 was reported previously (Ishijima et. al. Sci Rep 2017). Please compare them and discuss the relevance.

Line 321. It is vague conclusion.

Fig 1:

The resolution is too low. Please replace it.

Number of isolates would be better rather than number of cases

Please mention the definition of “patient serotype” and “non-patient serotype” in the legend.

Table S1. It seems that this table is identical to Table 1.

Table S1 and S2.

Table title should be included.

Fig. S1

Stx2e positive O157 would be rare. Did you confirm the result by PCR or other methods?

Reviewer #2: The manuscript, Genomic analysis of Shiga toxin-producing E. coli from patients and asymptomatic food handlers in Japan, written by Baba et al is a good manuscript that will enrich our understanding of molecular epidemiology of STEC and correlation of STEC, virulence factors, and HUS.

I have the following comments:

1- Page 18 Line 247: please change the word form to from (the isolates form patients).

2- Page 18 Lines 248-249: reword the following sentence because it is not clear what you mean by that: were significantly more often present among frequent among non-O157 isolates than O157.

3- Add bootstrap analysis to the phylogenetic tree.

4- In Materials and methods, WGS section: can you add the criteria for accepting a WGS run.

5- In genetic analysis section: Can you please elaborate on genome assembly. How did you trim the reads, does Platanus assembler do de novo assembly, what kind of software is used for this assembly (spades, shovil, or velvet), did you filter out small contigs? what sizes did you filter out if any?

6. PLOS authors have the option to publish the peer review history of their article (what does this mean?). If published, this will include your full peer review and any attached files.

Reviewer #1: No

Reviewer #2: No

---

## [Author Response · Author response to Decision Letter 0]

18 Oct 2019

Dr. Joerg Heber

Editor-in-Chief of PLOS ONE

Re: Manuscript ID: PONE-D-19-20606

Dear Dr. Heber,

We thank you very much for evaluating our manuscript entitled “Genomic analysis of Shiga toxin-producing Escherichia coli from patients and asymptomatic food handlers in Japan” (PONE-D-19-20606) and inviting a revision. We are pleased that the reviewers found our study interesting and are grateful for their thoughtful comments.

We have addressed each of the editor and the reviewers’ concerns in a point-by-point response, with each comment and response numbered sequentially as C and R. Each comment is stated in bold font, and our response is in standard font.

We again thank the reviewers for their careful review of our manuscript and hope that the manuscript is now acceptable for publication in PLOS ONE.

Hiroaki Baba, M.D., PhD

Department of Infectious Diseases, Tohoku University Graduate School of Medicine

1-1 Seiryo-machi, Aoba-ku, Sendai, Miyagi Prefecture 980-8574, Japan

TEL: 81-22-717-7373

E-mail: hbaba48@med.tohoku.ac.jp

Our responses to the comments from the Editor:

C1-1 Line 69: name the location in Japan

R1-1 We thank the editor for this comment. We have added “Miyagi Prefecture” on p.5, line 72.

C1-2 Lines 78-80: Where were the patient fecal samples obtained (e.g., hospitals?) and how were the asymptomatic individuals selected? Please clarify in the text.

R1-2 We agree that these points require clarification. We obtained fecal samples of hospital patients, so we have added “Thirty-eight isolates were obtained from fecal samples of hospital patients” on p.6, lines 82-83. Regarding the asymptomatic individuals, we collected all the STEC isolates detected through public health surveillance. We have clarified this on p.6, lines 79-81.

C1-3 Line 91: Bacterial DNA from the 65 isolates was extracted….

R1-3 This change has been made.

C1-4 Line 179: Again, here the phrase “isolates with patient serotypes”is awkward/unclear.

R1-4 We agree with your assessment. Since the terms “patient serotype” and “non-patient serotype” were confusing, we have replaced these terms throughout the paper with “major serotype” and “minor serotype”, respectively. The terms “major serotypes” and “minor serotypes” were defined as “serotypes found in patients in this study” and “serotypes only found in food handlers in this study” (Please see p.11, lines 161-165).

C1-5 Line 180: What is meant by “detected in 41-47 isolates”?

R1-5 We thank the editor for this comment. We have changed this sentence to “Among the 48 isolates with pathogenic serotypes, adhesion genes eae, tir, and espB, and secretion system genes espA, espJ, nleA, nleB, and nleC were detected in 41/48 (85%) isolates to 47/48 (98%) isolates” (p.14, lines 206-210).

C1-6 Lines 183-184: Consider changing this title to read more clearly.

R1-6 Please see R1-4 above.

C1-7 Line 242: Give examples/information in the text on which highly virulent STEC showed clonal expansion.

R1-7 We have added an example for highly virulent STEC as follows: “WGS analysis revealed clonal expansion of highly virulent STEC strains (e.g., O26:H11 strain with stx1a+stx2a) in our region” on p.20, lines 278-280.

C1-8 Please make the following these changes in the text: 

1. Line 21: ...However, the molecular…

2. Line 23: ...and asymptomatic food handlers…

3. Line 30: Delete “which”

4. Line 39: …genes were…

5. Line 49: …after a massive…

6. Line 58: …In addition, highly pathogenic STEC contain…

7. Line 64: …a comprehensive…

8. Line 94: MiSeq…300-bp reads, …

9. Line 146: …suggesting a close relationship between…

10. Line 151: …of the remaining…

11. Line 152: …whereas the ages of the food handlers…

12. Line 188: …had a comparable…

13. Lines 192-193: Should this read…often found in isolates from patients that were infants and small children…”

14. Line 194: …from an HUS patient…

15. Line 205: …with patient serotypes…

16. Line 233: …patients younger than 5 years of age.

17. Line 247: …isolates from patients,….

18. Line 249: …present among non-O157 isolates than from O157.

19. Line 270: The majority (17…

20. Line 271: …low virulence non-patient serotypes.

21. Line 272: …focused on patient-related serotypes…

22. Line 276: …two clusters of serogroup O103 had different…

23. Line 279: …O-antigen gene…

24. Line 292: …antibiotics based on the One…

25. Line 297: …cluster, and the O26:H11 isolates…

R1-8 We thank the editor for pointing out grammatical and typographical errors and have corrected these throughout the manuscript.

Our responses to the comments from Reviewer 1:

In this study, the authors revealed the characteristics and phylogeny of STEC in genomic level. As scarce information is available for Japanese STEC isolates, the information is useful. However, some of the discussion is superficial and the authors should consider the results in deeper depth. For instance, when the results were compared to the previous surveillance results of Japan or other countries, what is the implication?

In addition, some sentences are difficult to intuitively understand. English editing is recommended if it was not done yet.

We appreciate the reviewer’s comment on this point. To the best of our knowledge, this is the first genomic epidemiological study to investigate STEC isolates from asymptomatic food handlers. In Japan, STEC surveillance was instituted for patients and asymptomatic food handlers after a massive STEC epidemic associated with consumption of white radish sprouts occurred in 1996, and it has been found that the incidence of asymptomatic carriage, estimated as 84.2/100,000 population [Morita-Ishihara T, et al. Emerg Infect Dis 2016], is much higher than that of STEC infection. The molecular epidemiology of STEC isolates from food handlers, however, is not entirely known. Our study revealed that isolates with 5 non-O157 serotypes (O26:H11, O103:H2, O103:H8, O121:H19, and O145:H28) which had as many virulence genes as O157 isolates were prevalent among both patients and food handlers. Apart from O103:H8, these serotypes have been linked to epidemics and serious infections and have been frequently detected among clinical isolates worldwide. In contrast, isolates with remaining 12 serotypes which were only found in food handlers in this study were negative for major virulence genes, eae, tir, espB, espA, espJ, nleA, nleB and nleC. 

WGS analysis revealed clonal expansion of highly virulent STEC strains (e.g., O26:H11 strain with stx1a+stx2a) among both patients and food handlers in Miyagi prefecture, Japan. These findings demonstrated the importance of monitoring the genomic characteristics of STEC isolates from asymptomatic food handlers in addition to symptomatic patients.

This manuscript has been edited and rewritten by an experienced scientific editor, who has improved the grammar and stylistic expression of the manuscript.

C2-1 Line 23. our region -> Miyagi prefecture, Japan

R2-1 This change has been made.

C2-2 Line 26. What “n=19, 29%” meant was unclear. There are 20 isolates of O157 and O26 according to Fig. S1.

R2-2 We think the reviewer is mistaken on this point. Perhaps our explanation was not clear enough. There were 20 isolates of O157 and 20 isolates of O26. While the O157 isolates belonged to O157:H7 ST11 and ST2966, the O26 isolates belonged to O26:H11 ST21 and ST1705. Thus, there were 19 isolates of O157:H7 ST11 and 19 isolates of O26:H11 ST21.

C2-3 Line 29. Add “type III” before “secretion system”.

R1-3 We have added it in accordance with this comment.

C2-4 Line 76. remove “During a one-year period”.

R1-4 We have removed this phrase.

C2-5 Line 76. How were strains isolated? Please add the details of isolation.

R2-5 We agree that this point requires clarification, and we have added the following text to the “Material and Methods” section (p. 6, lines 84-88): Isolation of STEC from stool samples was done with sorbitol-MacConkey agar containing cefixime and tellurite, in addition to conventional E. coli isolation agar (e.g., triple sugar iron agar and lysine-indole-motility medium). A latex agglutination test (VTEC-RPLA, Denka Seiken, Japan) and PCR with the EVT-1&2 and EVS-1&2 primers (TaKaRa Biomedicals, Tokyo, Japan) were used to detect Stx and Stx genes, respectively.

C2-6 Are there any epidemiological links between the isolates used in this study? Discuss it in the Discussion section.

R2-6 We thank the reviewer for this comment. All 65 isolates analyzed in this study were not epidemiologically linked. We have added this information to the “Material and Methods” section (p.6, lines 79-81).

C2-7 Were all the strains isolated in Miyagi prefecture during the study period included in this study?

R2-7 All isolates from Miyagi Prefecture during the study period were included in this study. We have added this point on p.6, lines 79-81.

C2-8 In this study, serotypes of the isolates were mentioned. However, it is not mentioned how they were determined. For instance, what did OUT or H- mean. No identical O genotype even in in silico analysis? Did H- mean “no motility”? If so, did you search fliC genotype?

R2-8 We used CGE SerotypeFinder 1.1 to identify serotypes (Please see p.8, lines 115).

Since “O-” was unclear, we have changed O- to OUT (O-serotype untypable) (Please see Fig 1, Fig 2, and S1 Fig). 

No identical O serotype was detected in the one serotype OUT isolate in this study (isolate ID: E58) by SerotypeFinder. In this isolate, no O-processing genes (wzx, wzy, wzm, and wzt) were detected by BLAST search (http://blast.ncbi.nlm.nih.gov). This isolate may be assigned to O14 or O57, since O-processing genes for O14 and O57 could not be identified in their genomes [DebRoy C, et al. PLoS One. 2016;11: e0147434]. We have added this information to the “Results” section (p.10, lines 143-147).

O145:H- isolates were assigned to O145:H28 after reanalysis with SerotypeFinder. We have corrected all relevant parts of the manuscript.

C2-9 Line 94. Add reagent kit name.

R2-9 We have mentioned the reagent kit name, NEBNext Ultra DNA Library Prep Kit for Illumina, in the “Material and Methods” section (p.7, lines 99-101).

C2-10 Line 97. It seems that the data is not published yet. Please make the data public when the manuscript is submitted.

R2-10 We have made these data public.

C2-11 Line 106. It is useful to search saa sequence, which is the major adhesin in LEE negative STEC. In Virulence Finder, saa cannot be detected, because almost half of the sequence is repeat sequences.

R2-11 We thank the reviewer for this suggestion. We searched for the major adhesin gene saa, which cannot be detected by VirulenceFinder, using BLAST (http://blast.ncbi.nlm.nih.gov). We have added this information to p.8, lines 117-119. The results of searching for saa showed that only 5 of 65 isolates (7%) possessed this gene. All of these isolates were non-O157 and were from food handlers. We have added these points to P.17, lines 227-228. These findings suggested that saa may not play an independent role in STEC pathogenicity in the isolates we investigated.

C2-12 Line 108. What strain was used as reference?

R2-12 We used the following 6 reference genomes: STEC O157:H7 Sakai (GenBank accession numbers: BA000007), STEC O157:H7 strain EDL933 (AE005174), STEC 157:H7 strain EC4115 (CP001164), STEC O157:H7 strain TW14359 (CP001368), STEC O26:H11 strain 11368 (AP010953), and STEC O103:H2 strain 12009 (AP010958). We have mentioned this point in the “Materials and Methods” section (p.8, lines 120-126).

C2-13 Line 115. Please add the citation of RAxML, the model used, and the number of bootstrap replicates.

Did you remove recombinogenic region?

R2-13 We agree that these points require clarification. We have added the version and citation of RAxML, and the number of boostrap replicates to the “Material and Methods” section (p.9, lines 128-130) as follows: “A phylogenic tree with 1000 boostrap replicates was constructed by using the randomized accelerated maximum likelihood (RAxML) program v 8.2.12 (https://cme.h-its.org/exelixis/web/software/raxml/index.html)”. 

Recombinogenic regions were removed using Parsnp v 1.2 (https://harvest.readthedocs.io/en/latest/content/parsnp/quickstart.html#advanced-usage). We have also added the following information to the “Material and Methods” section (p.8, line 120-122).

C2-14 Line 127. What does “new” mean in this sentence? For instance, several strains of ST2836 of O103:H8 can be found in EnteroBase.

R2-14 We are sorry for not providing an explanation. Escherichia coli strains of these 13 sequence types are found in EnteroBase, but these sequence types were new as STEC strains. We have added the following text to the “Results” section (p.10, lines142-144): Seven sequence types have already been reported as STECs causing human disease in Enterobase (http://enterobase.warwick.ac.uk), while the other 13 sequence types, including O103:H8 ST2836, are new as STEC strains.

C2-15 Line 142. According to the surveillance information of Japan, the “non-patient serotypes” in this study have been reported from the patients.

R2-15 In this study, “non-patient serotype” was defined as “serotypes only found in food handlers in this study”. Thus, isolates with non-patient serotypes could be found in patients elsewhere.

Since the terms “patient serotype” and “non-patient serotype” were confusing, we have replaced these terms throughout the paper with “major serotype” and “minor serotype”, respectively. Please see our response to the editor (R1-4 above) regarding the same comment.

C2-16 Line 146. How many SNPs are there in the different allele?

R2-16 We thank the reviewer for this comment. We have added following text to the “Results” section (p.11, lines 169-171): There was only one SNP difference between ST11 and ST2966 in purA, as well as between ST21 and ST1705 in gyrA.

C2-17 Line 158-160. Are they multiple comparison? If so, how was the P values were adjusted?

R2-17 These are not multiple comparisons. We compared the incidence in infants and small children with that in each of the other age groups (one by one) using Fisher’s exact test.

C2-18 Line 199. Please remove “and all 18 isolates had at least one AMR gene”, because it is redundant.

R2-18 We agreed with your assessment, and we have removed this text.

C2-19 Line 214. Is O26:H11 correct, rather than O157:H7?

R2-19 We appreciate this comment. We have changed this sentence as follows: “Within the O157:H7 cluster, isolates positive for stx1a+stx2a formed a subcluster. Isolates positive for microcin genes mcmA, mchB, mchC, and mchF were assigned to a subcluster within the O26:H11 cluster” (p.18, lines 249-251).

C2-20 Line 214. The O26 cluster in unclear in Fig. 1.

R2-20 We agree with your assessment. Accordingly, we have replaced the phylogenetic tree with a new one that has a clearer branch (see revised Fig 1).

C2-21 L230-233. The meaning of these sentences is unclear. The authors should clarify the implication of the information.

R1-21 We agreed with your assessment. We have changed this sentence as follows: “Among these 9 core genomes, 5 were from patients, 3 were from food handlers, and one was isolated from both a patient and a food handler” (p.18, lines 266-268).

C2-22 Line 240. What are major virulence genes? LEE-related genes?

R1-22 We have added “eae, tir, espB, espA, espJ, nleA, nleB, and nleC” to this section (p.20, line 278).

C2-23 Line 241. The etiologic agent of most of the young children patients was O26. Therefore, the serotype should be focused on.

R2-23 We appreciate the reviewer’s concern on this point. Among the isolates from patients, isolates with O103 and O121 were more likely to be from young children (both 1/2, 50%) than isolates with O157 (2/19, 11%), even though the total number of isolates was low. O26 isolates from patients and non-O157 isolates (excluding O26 isolates) from patients harbored an equal number of virulence genes (both had a median of 18 virulence genes per isolate, P=0.84). The distribution of putative virulence genes among was similar O26 isolates from patients and non-O157 isolates (excluding O26 isolates) from patients (Please see Table 1 in "Response to Reviewers"). 

The incidence of O26 infection was significantly higher in infants and small children (9.0) than in other age groups (all P<0.001), while there were no significant differences in the incidence of O157 infections among age groups (Please see Table 2 in "Response to Reviewers"). Stx gene stx1a, adhesion gene efa1, secretion system genes espF and cif, and fimbrial gene lpfA were significantly more frequent among O26 isolates from patients than among O157 isolates from patients (Please see Table 3 in "Response to Reviewers"). These findings were similar to the results obtained by analysis focusing on all non-O157 serotypes (including O26) from patients. Therefore, we would like to retain the original text in this section. 

C2-24 Line 254-256. These are paradoxical statements. The fact “women are more likely to work in food related occupations” do not explain that more than half of the cases of food handlers are women.

R2-24 We thank the reviewer for this comment. According to a survey by the Japanese Ministry of Economy, Trade and Industry (https://www.meti.go.jp/statistics/tyo/syokozi/result-2/h2c5kjaj.html), more than half of food handlers in Japan are women, and we think the predominance of women among STEC carriers may reflect this fact.

Because this point is not important, we have deleted the relevant text.

C2-25 Line 272. This suggestion is unclear. Please specify the recommendation (e.g. isolation method).

In Japan (and other countries), “non-patient serotypes” in this study are responsible for severe diseases, although the incidence rate is not high. How do the authors evaluate them?

R2-25 We thank the reviewer for raising these important points. The current STEC surveillance system for food handlers in Japan is only based on serotyping and detection of Stx. We think STEC surveillance should focus on major virulence genes, such as eae, tir, espB, espA, espJ, nleA, nleB, and nleC. We have modified our suggestion to clarify it (p.22, lines 319-322). 

As we mentioned on p.22, lines 305-310, STEC are E. coli strains of different lineages that have acquired virulence genes independently at different time points, and STEC strains from the same O serogroups are polyphyletic since horizontal transfer of the O- antigen gene can occur among different E. coli strains. These points raise the possibility that new serotypes of highly virulent STEC strains may emerge and that highly virulent serotypes could differ from country to country.

C2-26 Line 290. “under certain conditions” should be clarified.

R2-26 We thank the reviewer for this suggestion. We have changed this text to: “Antimicrobial therapy is generally not recommended for STEC infection due to the possible risk of HUS, but it may be beneficial for patients with persistent diarrhea or food handlers with long-term STEC carriage” (p.23, lines 332-333).

C2-27 Line 294-298. Subclusters of O157 is also unclear in Fig. 1. What about clade of O157?

R2-27 Please see R2-20 above and the revised Fig 1. 

C2-28 Line 297-301. Virulence of O26:H11 was reported previously (Ishijima et. al. Sci Rep 2017). Please compare them and discuss the relevance.

R2-28 We appreciate this comment. Ishijima et al. reported a newly emerging virulent O26:H11/H- ST29 STEC clade in Japan. However, the O26:H11 strain with stx1a+stx2a in this study was assigned to a different sequence type, ST21. We have added this explanation to the “Discussion” section (p.24, lines 347-349).

C2-29 Line 321. It is vague conclusion.

R2-29 We agree with your assessment. We have added information about specific serotypes, virulence genes, and AMR genes to the conclusion (p.25, lines 364-368).

C2-30 Fig 1: The resolution is too low. Please replace it.

Number of isolates would be better rather than number of cases.

Please mention the definition of “patient serotype” and “non-patient serotype” in the legend.

R2-30 We thank the reviewer for these comments. We have replaced Fig 1 with a clearer one. We have changed “number of cases” to “number of isolates”. Instead of “patient” and “non-patient” serotypes, we have used the terms “major serotypes” and “minor serotypes”, which are defined as “serotypes found in patients in this study” and “serotypes only found in isolates from food handlers in this study”, respectively.

C2-31 Table S1. It seems that this table is identical to Table 1.

R2-31 We thank the reviewer for pointing out this error. We have replaced Table S1.

C2-32 Table S1 and S2. Table title should be included.

R2-32 Please see p.33, lines 497-498 and 502-504.

C2-33 Fig. S1. Stx2e positive O157 would be rare. Did you confirm the result by PCR or other methods?

R2-33 We apologize for this error. This strain was actually Stx2a positive. We have corrected this error throughout the manuscript.

Our responses to the comments from Reviewer 2:

The manuscript, Genomic analysis of Shiga toxin-producing E. coli from patients and asymptomatic food handlers in Japan, written by Baba et al is a good manuscript that will enrich our understanding of molecular epidemiology of STEC and correlation of STEC, virulence factors, and HUS.

C3-1 Page 18 Line 247: please change the word form to from (the isolates form patients).

R3-1 This change has been made.

C3-2 Page 18 Lines 248-249: reword the following sentence because it is not clear what you mean by that: were significantly more often present among frequent among non-O157 isolates than O157.

R3-2 We thank the reviewer for pointing out this error and we have corrected this sentence by removing “often present among” (p.21, line 288).

C3-3 Add bootstrap analysis to the phylogenetic tree.

R3-3 Please see our response to reviewer 1 (R2-13 above) regarding the same comment.

C3-4 In Materials and methods, WGS section: can you add the criteria for accepting a WGS run.

R3-4 We agree with your point and we have added the following text to Material and Methods (p.7, lines 104-105): “The passing filter ranged from 90.88 to 96.74% (mean, 93.19%), and the average Q30 ranged from 78.30 to 87.21% (mean, 83.50%)”.

C3-5 In genetic analysis section: Can you please elaborate on genome assembly. How did you trim the reads, does Platanus assembler do de novo assembly, what kind of software is used for this assembly (spades, shovil, or velvet), did you filter out small contigs? what sizes did you filter out if any?

R3-5 We thank for raising this important question. We have added the following explanation: “Sequence reads were trimmed of adapters and filtered to remove reads shorter than 36 bp using Trimmomatic, followed by assembly using Platanus assembler v 1.2.4.” (p.8, lines 110-111).

---

## [Editor Report · Decision Letter 1]

30 Oct 2019

PONE-D-19-20606R1

Genomic analysis of Shiga toxin-producing Escherichia coli from patients and asymptomatic food handlers in Japan

PLOS ONE

Dear Dr. Baba,

Thank you for submitting your manuscript to PLOS ONE. After careful consideration, we feel that it has merit but does not fully meet PLOS ONE’s publication criteria as it currently stands. Therefore, we invite you to submit a revised version of the manuscript that addresses the points raised during the review process.

The revised manuscript is improved; however, additional changes that are needed are the following:

Line 20:  comma after disease.

Line 29:  comma after tir.

Line 30: comma after nleB.

Line 44:  change causes to cause.

Line 46:  change to -  …..serotype, severe infections caused by non-O157 serogroups are…..

Line 64:  change to ….pathogenicity and antibiotic resistance….

Line 118: change to ….. (AE005174), STEC O157:H7…..

Line 122: change to ….A phylogenetic tree….

Line 124: change to ……and it was visualized…..

Lines 137-141:  change to …….STEC strains.  An O-group was not detected in the one OUT (O-serogroup untypable) isolate by SerotypeFinder.  In this isolate, no O-processing genes (wzx, wzy, wzm, wzt) were detected by BLAST.  It is possible that this isolate could be assigned to serogroups O14 or O57 since O-processing genes for these serogroups have not been found in their genomes [20], or it could represent a new serogroup.

Line 157:  In Figure 1, O145:H28 is grouped as a minor serotype, not a major serotype.  Please clarify/correct.

Lines 194-197:  This sentence is still not clear as written.  To what do the figures 41/48 and 47/48 refer (which genes or groups of genes)? Please rewrite this sentence based on what is shown in Table 2.   

Line 323:  change to ……to provide a better……..

Line 347:  Int J Infect Dis.

Line 351:  italicize gene name.

References #13, 20, 25, 35, 38 – The first letter of the words in the title of the paper should not be capitalized.

References #20 and 35 – italicize Escherichia coli.

Line 394:  Is this written correctly?

Line 417:  italicize stx

Line 418:  Front Cell Infect Microbiol.

Line 435:  Euro Surveill.

We would appreciate receiving your revised manuscript by Dec 14 2019 11:59PM. To enhance the reproducibility of your results, we recommend that if applicable you deposit your laboratory protocols in protocols.io, where a protocol can be assigned its own identifier (DOI) such that it can be cited independently in the future. For instructions see: http://journals.plos.org/plosone/s/submission-guidelines#loc-laboratory-protocols

We look forward to receiving your revised manuscript.

Kind regards,

Pina Fratamico, Ph.D.

Academic Editor

PLOS ONE

---

## [Author Response · Author response to Decision Letter 1]

31 Oct 2019

Our responses to the comments from the Editors:

The revised manuscript is improved; however, additional changes that are needed are the following:

C-1

Line 20: comma after disease.

Line 29: comma after tir.

Line 30: comma after nleB.

Line 44: change causes to cause.

Line 46: change to - …..serotype, severe infections caused by non-O157 serogroups are…..

Line 64: change to ….pathogenicity and antibiotic resistance….

Line 118: change to ….. (AE005174), STEC O157:H7…..

Line 122: change to ….A phylogenetic tree….

Line 124: change to ……and it was visualized…..

Lines 137-141: change to …….STEC strains. An O-group was not detected in the one OUT (O-serogroup untypable) isolate by SerotypeFinder. In this isolate, no O-processing genes (wzx, wzy, wzm, wzt) were detected by BLAST. It is possible that this isolate could be assigned to serogroups O14 or O57 since O-processing genes for these serogroups have not been found in their genomes [20], or it could represent a new serogroup.

Line 323: change to ……to provide a better……..

Line 347: Int J Infect Dis.

Line 351: italicize gene name.

References #13, 20, 25, 35, 38 – The first letter of the words in the title of the paper should not be capitalized.

References #20 and 35 – italicize Escherichia coli.

Line 417: italicize stx

Line 418: Front Cell Infect Microbiol.

Line 435: Euro Surveill.

R-1 We thank the editor for pointing out grammatical and typographical errors and have corrected these throughout the manuscript.

C-2 Line 157: In Figure 1, O145:H28 is grouped as a minor serotype, not a major serotype. Please clarify/correct.

R-2 We thank the editor for pointing out this error. We grouped O145:H28 as a major serotype in Figure 1 (Please see revised Figure 1).

C-3 Lines 194-197: This sentence is still not clear as written. To what do the figures 41/48 and 47/48 refer (which genes or groups of genes)? Please rewrite this sentence based on what is shown in Table 2.

R-3 We thank the editor for this comment. We have changed this sentence to “Among the 48 isolates with major serotypes, adhesion genes eae, tir, and espB were detected in 47 (98%), 46 (96%), and 45 (94%) isolates respectively, and secretion system genes espA, espJ, nleA, nleB, and nleC were detected in 46 (96%), 46 (96%), 41 (85%), 45 (94%), and 41 (85%) isolates respectively” (p.14, lines 197-200).

C-4 Line 394: Is this written correctly?

R-4 We thank the editor for this comment. We have changed reference #19 to “Stamatakis A. Using RAxML to infer phylogenies. Curr Protoc Bioinformatics. 2015;51: 6.14.1-14.

---

## [Editor Report · Decision Letter 2]

4 Nov 2019

Genomic analysis of Shiga toxin-producing Escherichia coli from patients and asymptomatic food handlers in Japan

PONE-D-19-20606R2

Dear Dr. Baba,

We are pleased to inform you that your manuscript has been judged scientifically suitable for publication and will be formally accepted for publication once it complies with all outstanding technical requirements.

With kind regards,

Pina Fratamico, Ph.D.

Academic Editor

PLOS ONE
---

## [Editor Report · Acceptance letter]

8 Nov 2019

PONE-D-19-20606R2 

Genomic analysis of Shiga toxin-producing *Escherichia coli* from patients and asymptomatic food handlers in Japan

Dear Dr. Baba:

I am pleased to inform you that your manuscript has been deemed suitable for publication in PLOS ONE. Congratulations! Your manuscript is now with our production department. 

With kind regards,

on behalf of

Dr Pina Fratamico 

Academic Editor

PLOS ONE